# Atomic Information Flow: A Network Flow Model for Tool Attributions in RAG Systems

## Abstract

Many tool-based Retrieval Augmented Generation (RAG) systems lack precise mechanisms for tracing final responses back to specific tool components—a critical gap as systems scale to complex multi-agent architectures. We present **Atomic Information Flow (AIF)**, a graph-based network flow model that decomposes tool outputs and LLM calls into atoms: indivisible, self-contained units of information. By modeling LLM orchestration as a directed flow of atoms from tool and LLM nodes to a response super-sink, AIF enables granular attribution metrics for AI explainability.

Motivated by the max-flow min-cut theorem in network flow theory, we train a lightweight Gemma3 (4B parameter) language model as a context compressor to approximate the minimum cut of tool atoms using flow signals computed offline by AIF. We note that the base Gemma3-4B model struggles to identify critical information with **54.7%** accuracy on HotpotQA, barely outperforming lexical baselines (BM25). However, post-training on AIF signals boosts accuracy to **82.71%** (+28.01 points) while achieving **87.52%** (+1.85%) context token compression—bridging the gap with the Gemma3-27B variant, a model nearly $7\times$ larger.

## 1. Introduction

The emergence of RAG (Lewis et al., 2021) (Gao et al., 2024) as a framework to introduce data from external databases into LLM generations has inspired powerful tool calling capabilities in modern large language models. The powerful capabilities that RAG introduced has also inspired directions in post training (Jin et al., 2025) and interesting multi-agent orchestration designs (Dang et al., 2025) to improve the ability for systems to optimally leverage external retriever systems.

Multi-tool RAG systems are commonly modeled as a graph,

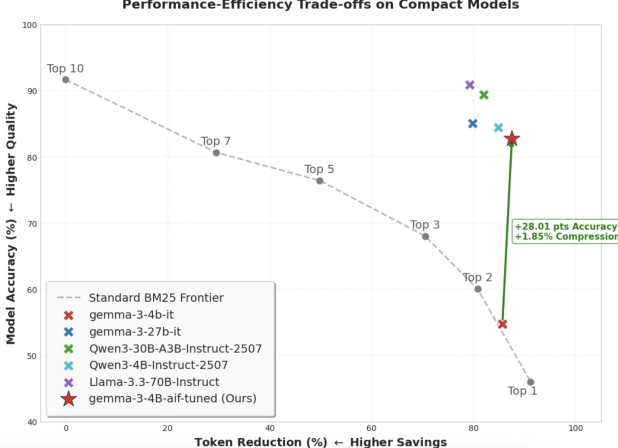

*Figure 1.* Minimum Cut Signals Derived from AIF significantly outperform the base model and lexical baselines on HotPotQA, bridging the gap against much larger model architectures

where tools are treated as nodes in the orchestration path for trajectory level optimizations (Wu et al., 2025). Inspired by this idea, we look towards graph network flow (FORD & Fulkerson, 1956) as a model for information propagation through RAG systems. We call this model **Atomic Information Flow (AIF)**. In AIF, the query, tool calls, LLM calls, and response in the RAG system are defined as nodes. Edges represent the sequencing of the nodes in a directed manner, where the query is the super-source and the response is the super sink.

More concretely, we define self contained snippets of information as **atoms**. Every tool output and LLM generation within a RAG system is composed of these atoms. In *Atomic Information Flow*, tool outputs and LLM calls are modeled as atom supply nodes. The *query* is modeled as a super-source node as it is the entry point to the RAG system, while the final LLM generated *response* is modeled as a super-sink node, given that is is the terminal state of the RAG system. We define **flow** as the directed movement of atoms along the edges in the RAG system.

In RAG systems, a single user query can trigger a cas-

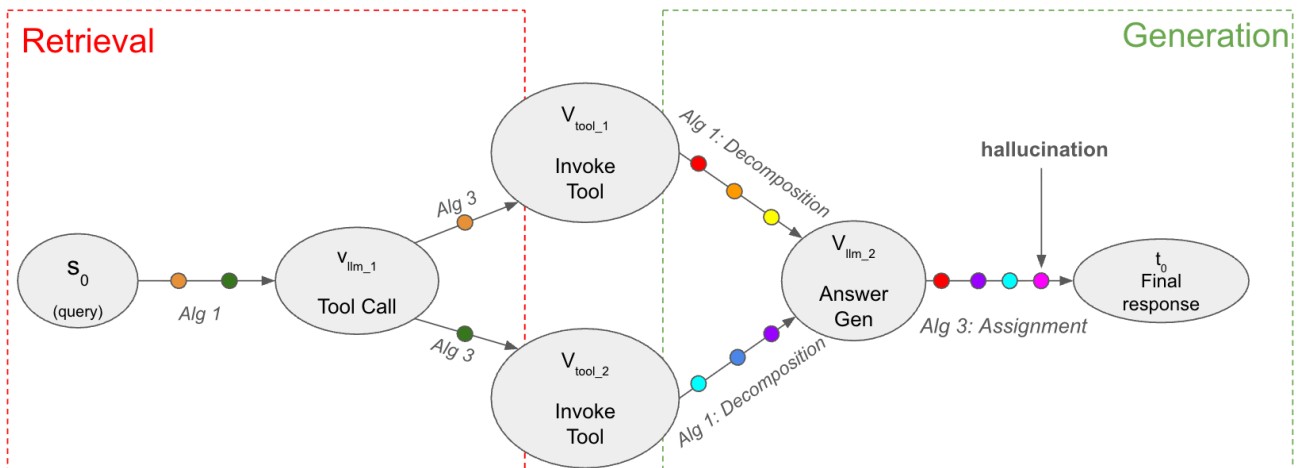

*Figure 2.* The AIF model. Dots of the same color denote atoms that flow through each LLM "gate". For the scope of this paper, we focus on the Generation component and leave the Retrieval flow edges for future work. See section 8.1 for more details. Details on decomposition and assignment algorithms can be found at Alg 1 and Alg 3

cade of actions: multiple retriever calls, SQL queries, web searches, and several intermediate LLM calls that rewrite or summarize partial results. When something goes wrong—hallucinated facts, missing evidence, or inefficient tool use—it is extremely difficult to answer questions like:

- Which tool outputs actually influenced this specific sentence in the final answer?

- Which tools could we have safely skipped without changing the answer?

- Where is the true information bottleneck that determines answer quality?

AIF is designed exactly to answer these questions. By decomposing outputs into atoms and tracking their flow through the orchestration graph, AIF provides:

- **Fine-grained tool attribution:** how much of the final answer is supported by which tools.

- **Trajectory debugging:** identifying tools or calls that introduce irrelevant or misleading information.

- **Compression signals:** a principled way to locate the minimum information cut—the smallest set of atoms needed to preserve answer fidelity—enabling us to train compact context-compressor models.

## 2. The Atomic Information Flow Model

We introduce the core formal objects used in our Atomic Information Flow (AIF) framework, as they appear in Figure 2.

**Definition 2.1** (Graph). A Retrieval-Augmented Generation (RAG) system is modeled as a directed graph $G = (V, E)$ where $V$ is the set of nodes, and $E \subseteq V \times V$ is the set of directed edges representing causal or sequential dependencies between nodes (Wu et al., 2025).

*Remark* 2.2. While we define nodes as "tools" in this paper for simplicity, the graph model may generalize tool nodes to agent nodes in deeper multi-agent systems (MAS) with sub-tools.

**Definition 2.3** (Atom). Let $\mathcal{A}$ denote the universe of possible information units. An *atom* is a minimal, self-contained unit of semantic information used for attribution and flow analysis (Srikanth & Rudinger, 2025; FORD & Fulkerson, 1956). For each node $v \in V$, let $\mathsf{Atoms}(v) \subseteq \mathcal{A}$ denote the multiset of atoms produced at node $v$.

**Definition 2.4** (Source/Supply Node). A node $v \in V$ is a *supply node* if it introduces new atoms into the system:

$$\mathsf{Atoms}(v) \neq \emptyset.$$

Typical supply nodes correspond to tool invocations or intermediate LLM generations.

**Definition 2.5** (Super-Source). The *super-source*, denoted $s_0$, represents the user query and provides the initial set of atoms:

$$\mathsf{Atoms}(s_0) = \mathsf{Atoms}(\mathsf{query}).$$

By construction, $s_0$ has no incoming edges:

$$\forall u \in V, \ (u, s_0) \notin E.$$

**Definition 2.6** (Super-Sink). The *super-sink*, denoted $t_0$, represents the final LLM response. It consumes atoms and

has no outgoing edges:

$$\forall v \in V, \ (t_0, v) \notin E.$$

In addition, no atoms originate at the super-sink:

$$\mathsf{Atoms}(t_0) = \emptyset.$$

**Definition 2.7** (Node Typing). In multi-tool RAG settings, the node set decomposes as:

$$V \ = \ \{s_0\} \ \cup \ V_{\text{tool}} \ \cup \ V_{\text{llm}} \ \cup \ \{t_0\},$$

where $s_0$ is the super-source (query), $V_{\text{tool}}$ are tool call nodes, $V_{\text{llm}}$ are LLM call nodes, and $t_0$ is the super-sink (final response).

**Definition 2.8** (Flow with Supply). A *flow* is a function $f : E \to \mathbb{N}_0$ assigning a non-negative number of atoms to each directed edge. Let $s : V \to \mathbb{N}_0$ be a *supply function* defined by $s(v) = |\mathsf{Atoms}(v)|$, representing the count of atoms introduced externally at node $v$. For any non-terminal node $v \in V \setminus \{s_0, t_0\}$, the flow satisfies the *relaxed* conservation law:

$$\sum_{(u,v) \in E} f(u,v) + s(v) \ \geq \ \sum_{(v,w) \in E} f(v,w).$$

*Remark* 2.9 (Active Nodes & Steering). Unlike passive transport nodes, we model LLM nodes as active components that perform both *amplification* (via $s(v)$) and *filtering*. In linear programming terms, the inequality implies the existence of a non-negative *slack variable* $\delta(v)$ such that $\sum f_{\text{in}} + s(v) = \sum f_{\text{out}} + \delta(v)$. This slack $\delta(v)$ represents the volume of irrelevant atoms *gated* (discarded) by the LLM's steering instructions.

*Remark* 2.10 (Multicommodity Decomposition). While we denote flow $f(u,v)$ as a scalar representing total information volume, strictly speaking, this represents a *Semantic Multicommodity Flow*. The edge $(u,v)$ serves as a channel for multiple distinct, non-fungible semantic units, such that $f(u,v) = \sum_{a \in \mathcal{A}} \mathbb{I}[\text{atom } a \text{ traverses } (u,v)]$.

This formulation maps the atomic flow maximization problem to Integer Multicommodity Flow (Even et al., 1976), which is NP-hard. Leveraging the *max-flow min-cut duality* (FORD & Fulkerson, 1956), we employ a learned policy $\pi(q, T)$ in section 7 via SFT (Guo et al., 2025) to approximate the optimal cut—equivalent to identifying the information bottleneck—rather than solving for the exact flow analytically at inference time.

## 3. Related Work

Our work builds on two major lines of research: **factual decomposition** and **source attribution**. *Atomic Information Flow* draws inspiration from both, advancing toward

a unified framework that integrates and generalizes these well-studied problems towards tool-based LLM systems.

**Factual Decomposition.** Decomposing text into minimal units is a proven strategy for enhancing faithfulness and interpretability. Prior work utilizes fact-level modeling to resolve conflicts in RAG (Zhang et al., 2025) and reveal logical inconsistencies in NLI (Srikanth & Rudinger, 2025). Granular decomposition further enables "Decompose-Verify-Revise" loops to reduce hallucinations (Yan et al., 2025; 202, 2025) and improves retrieval in multi-hop QA via question decomposition (Ammann et al., 2025; Tang & Yang, 2024). Through AIF, we naturally incorporate *FActScore* (Min et al., 2023) as the GROUNDEDNESS metric in Table 1, measuring the ratio of supported to total atomic facts:

$$\text{FActScore} = \frac{\#\text{ supported atomic facts}}{\#\text{ total atomic facts}}.$$

**Source Attribution.** Research on tracing generated content to sources has been standardized through benchmarks like *DeepResearch Bench* (Du et al., 2025) and *ALCE* (Gao et al., 2023), which evaluates citation fidelity using the AIS framework (Rashkin et al., 2023). Other approaches refine granularity through span-level attribution queries (Hirsch et al., 2025) or synthetic data pipelines for grounding (Radevski et al., 2025). AIF extends these methods by moving beyond external evidence alignment to **global provenance attribution**, tracing the origin and transformation of semantic atoms across the entire internal orchestration graph of tool-augmented systems.

## 4. Methodology

We model information flow by decomposing the RAG process into minimal semantic units (*atoms*). This pipeline consists of three stages: (1) atomic decomposition of tool outputs, (2) atomic signal injection, and (3) response atom assignment.

### 4.1. Stage 1: Atomic Decomposition

We first decompose tool outputs into atoms using a model $D$ (GPT5-Nano (minimal reasoning)) (Singh et al., 2025). To circumvent context window constraints on long outputs, we adopt a map–reduce strategy (Zhou et al., 2024) as needed: outputs are chunked, decomposed in parallel, and merged.

**Definition 4.1** (Atomic Decomposition). Given tool outputs $\{t_i\}_{i=1}^m$, the problem is to find, for each $t_i$, a set of spans $a_i = \{a_{i,1}, \ldots, a_{i,n_i}\}$ such that each $a_{i,j}$ is indivisible and their union $\bigcup a_i$ is informationally complete with respect to $t_i$.

**Algorithm 1** Atom Decomposition for Tool Calls

1: **Input:** Tool calls $T = (t_1, \ldots, t_m)$, Decomposer $D(\cdot)$
2: **Output:** Map $A : \{1 \ldots m\} \to \text{List(Atom)}$
3: $A \leftarrow \emptyset$
4: **for** $i \leftarrow 1$ **to** $m$ **do**
5:     $A[i] \leftarrow D(t_i)$    {Apply decomposition (with map-reduce if needed)}
6: **end for**
7: **return** $A$

## 4.2. Stage 2: Atomic Signal Injection

A core capability of the AIF framework is the modular injection of scalar metadata into the graph, allowing information flow to be modulated by auxiliary signals such as source authority, temporal freshness, or uncertainty. We formalize this process as *Atomic Signal Injection*.

For our experimental validation, we instantiate this mechanism using a semantic relevance scorer $S(\cdot)$ that assigns Likert-scale values relative to the query $q$. While we acknowledge the limitations of reference-free metrics in capturing ground truth (Girrbach et al., 2025; Liu et al., 2023)—and analyze the specific behavior of this relevance signal in Appendix G—we utilize this setup to demonstrate the framework's extensibility to arbitrary weighted flow heuristics.

**Definition 4.2** (Atomic Signal Injection). Given atoms $A_{all} = \bigcup A[t]$ and query $q$, Atomic Signal Injection applies a scoring policy $S(\cdot)$ to generate a signal map where every atom $a \in A_{all}$ is assigned a scalar metadata value $S(a, q) \in \mathcal{S}$.

**Algorithm 2** Atomic Signal Injection

1: **Input:** Tools $T$, Atom Map $A$ (from Alg. 1), Query $q$, Scorer $S(\cdot)$
2: **Output:** Signal Map $M : T \to \text{List}(\mathcal{S})$
3: **for** each $t \in T$ **do**
4:     $M[t] \leftarrow [\,]$
5:     **for** each atom $a \in A[t]$ **do**
6:         {Inject signal using policy $S$ (e.g., Relevance, Freshness)}
7:         $M[t].\text{append}(S(a, q))$
8:     **end for**
9: **end for**
10: **return** $M$

See Figure 3 for a concrete example of decomposed and labeled atoms.

*Figure 3.* Decomposition and relevance labeling for a HotpotQA tool passage. Full details in Appendix D.

## 4.3. Stage 3: Response Atom Assignment

Finally, we induce flow edges by mapping response atoms back to their source tool atoms. To ensure the matching process is unbiased by tool order or hierarchy, we flatten all tool atoms into a single global candidate list $U_{\text{flat}}$. Let $R_{resp} = (r_1, \ldots, r_k)$ be the atoms of the final LLM response. We determine the origin of each $r_j$ by selecting the best set of atoms from $U_{\text{flat}}$ and resolving the multi-set of source tools post-hoc. Note that we collect multiple atom attributions to capture potential multi-hop attributions to the response atom. We formally define the approach in algorithm 3

The matching function $M$ utilizes a map–reduce strategy over $U_{\text{flat}}$ for large contexts, ensuring that every candidate atom is considered regardless of its source tool position.

We define a set of **flow heuristics** in Table 1 to aggregate and quantify various aspects of the resulting AIF structure.

**Algorithm 3** Response Atom Assignment (Global Pool)

**Require:** Tool Atom Map $A : \{1..m\} \rightarrow \text{List(Atom)}$, Response Atoms $R_{resp}$, Matcher $M(\cdot)$

**Ensure:** Attribution Map $\Phi : R_{resp} \rightarrow \mathcal{P}(T)$ // Maps to a set of Tool IDs

1: // 1. Flatten all atoms into a global unbiased list
2: $U_{\text{flat}} \leftarrow [\ ]$
3: $S_{\text{map}} \leftarrow [\ ]$ // Tracks source tool ID for each atom index
4: **for** each tool ID $t \in \{1..m\}$ **do**
5:     **for** each atom $a \in A[t]$ **do**
6:         $\text{APPEND}(U_{\text{flat}}, a)$
7:         $\text{APPEND}(S_{\text{map}}, t)$
8:     **end for**
9: **end for**

10: // 2. Assign response atoms to global candidates (Multi-hop)
11: **for** each $r_j \in R_{resp}$ **do**
12:     $\Phi[r_j] \leftarrow \emptyset$
13:     $K \leftarrow M(r_j, U_{\text{flat}})$ // Returns set of indices $K$ matching $r_j$
14:     **if** $K \neq \emptyset$ **then**
15:         **for** each index $k \in K$ **do**
16:             $t_{id} \leftarrow S_{\text{map}}[k]$
17:             $\Phi[r_j] \leftarrow \Phi[r_j] \cup \{t_{id}\}$ // Accumulate sources for multi-hop
18:         **end for**
19:     **else**
20:         $\Phi[r_j] \leftarrow [\ ]$
21:     **end if**
22: **end for**
23: **return** $\Phi$

### 4.4. Flow Heuristics

We formally define a set of **flow heuristics** in Table 1 to aggregate and quantify various aspects of the AIF model.

We define the notation for our flow heuristics as follows: $A_R$ denotes the set of atoms in the final response; $A_T$ is the set of all atoms provided by tools; and $A_{T,t}$ is the set of atoms contributed by tool $t$. $A_{R,T}$ refers to the set of response atoms attributable to any tool, while $A_{R,T,t}$ is the subset of response atoms attributed to tool $t$.

To incorporate atom relevance, we employ the Atom Relevance Labeling algorithm (Algorithm 2) which assigns a relevance score to each atom. For a relevance threshold $K$, we define $A_{R,T,\geq K}$ and $A_{T,\geq K}$ as the sets of response and tool atoms, respectively, whose assigned relevance scores are at least $K$. Tool-specific high-relevance sets $A_{R,T,t,\geq K}$ and $A_{T,t,\geq K}$ are similarly defined for atoms from tool $t$ with relevance above the threshold.

```
Response: Here I Am Again is an album by American
country singer-songwriter Loretta Lynn, who had a career
of almost 60 years.
```

```
Response Atomic Assignment (Flow Edge): [
  {
    "Fact": "Here I Am Again is an album by American
country singer-songwriter Loretta Lynn",
    "Assignment": 12,
    "Relevance": 5,
    "Tool_Id": "Here I Am Again"
  },
  {
    "Fact": "Loretta Lynn, who had a career of almost 60
years.",
    "Assignment": 3,
    "Relevance": 5,
    "Tool_Id": "Loretta Lynn"
  }
]
```

*Figure 4.* Response Assignment Example from HotpotQA. *Assignment* field matches a corresponding *Index* from Appendix D

## 5. Experimental Setup

We conduct experiments on QA datasets containing multiple context passages and human-labeled ground truth attributions to evaluate the alignment of the Atomic Information Flow (AIF) model with human judgments. In our setup, we treat each context document as a single tool call—consistent with the AIF model definitions—and provide it alongside the query to an LLM (GPT4.1) to generate a response (Prompt E). To assess performance, we employ an LLM-as-a-judge (GPT4.1) (Zheng et al., 2023) to compare the generated response against the human-annotated ground truth answer (Prompt F). Based on this evaluation, we partition each dataset into **True** and **False** segments. We utilize the **True** segment to rigorously measure attribution accuracy against ground truth, while retaining the **False** segment to analyze correlations between flow heuristics 1 and answer correctness.

We then apply the factual decomposition and attribution methodology(4) to assign atomic facts, their relevance to the input query, and their attributions. From here, we can measure the attribution accuracy using the ground truth context attribution labels from the human annotated datasets. We use HotpotQA (Yang et al., 2018), MS MarcoV2 (Bajaj et al., 2018), Musique (Trivedi et al., 2022), and Wiki Multihop QA (Ho et al., 2020) as our QA datasets, as they all have ground truth tool attributions to measure the efficacy of our flow computations.

We compare against the standard Vanilla baseline established in the ALCE benchmark (Gao et al., 2023) with the GPT5-Nano (minimal reasoning) model (Singh et al., 2025), which prompts the model to generate answers and citations

*Table 1.* Flow Heuristics

| METRIC NAME | NOTATION | DESCRIPTION | FORMULA |
| --- | --- | --- | --- |
| GROUNDEDNESS | RAP | MEASURES THE FRACTION OF RESPONSE ATOMS FROM TOOLS | $\frac{|A_{R,T}|}{|A_R|}$ |
| GROUNDEDNESS @ K | $\text{RAP}_K$ | MEASURES THE FRACTION OF RESPONSE ATOMS FROM HIGH-RELEVANCE TOOL ATOMS | $\frac{|A_{R,T \geq K}|}{|A_R|}$ |
| TOOL CONSUMPTION | RAR | PROPORTION OF ALL TOOL ATOMS THAT ARE CONSUMED IN RESPONSE | $\frac{|A_{R,T}|}{|A_T|}$ |
| TOOL CONSUMPTION @ K | $\text{RAR}_K$ | PROPORTION OF ALL HIGH RELEVANCE TOOL ATOMS CONSUMED IN RESPONSE | $\frac{|A_{R,T, \geq K}|}{|A_{T \geq K}|}$ |
| TOOL CONTRIBUTION | $\text{TUP}_t$ | FOR TOOL $t$, MEASURES PROPORTION OF RESPONSE COMING FROM $t$ | $\frac{|A_{R,T,t}|}{|A_R|}$ |
| TOOL CONTRIBUTION @ K | $\text{TUP}_{t,K}$ | FOR TOOL $t$, MEASURES PROPORTION OF RESPONSE COMING FROM HIGH RELEVANCE ATOMS IN $t$ | $\frac{|A_{R,T,t \geq K}|}{|A_R|}$ |
| TOOL USAGE | $\text{TUR}_t$ | FOR TOOL $t$, MEASURES PROPORTION OF $t$ ATOMS CONSUMED IN THE RESPONSE | $\frac{|A_{R,T,t}|}{|A_{T,t}|}$ |
| TOOL USAGE @ K | $\text{TUR}_{t,K}$ | FOR TOOL $t$, MEASURES PROPORTION OF $t$'S HIGH-RELEVANCE ATOMS CONSUMED IN RESPONSE | $\frac{|A_{R,T,t \geq K}|}{|A_{T,t \geq K}|}$ |

in a single pass. We report our results in Table 2.

## 6. Results and Analysis

### 6.1. Tool Attribution on Benchmarks

Table 2 demonstrates that the **True** segments of each dataset exhibit significantly higher tool attribution precision and recall compared to the **False** segments. This performance gap indicates that AIF effectively captures the source grounding required for correct answers in these QA datasets.

Furthermore, AIF achieves attribution scores comparable to, and in some cases marginally exceeding, the ALCE baseline (Gao et al., 2023) (Table 2). This demonstrates the viability of our atomic decomposition approach: it matches the attribution performance of standard methods while unlocking the ability to compute granular, tool-level heuristics that single-shot citations cannot provide.

### 6.2. Human Annotation

To validate the reliability of our pipeline, we sampled 50 instances from HotPotQA and manually evaluated the accuracy of the Atomic Decomposition (Algorithm 1) and Response Atom Assignment (Algorithm 3) stages as a final sanity check. We annotated these stages in isolation to verify the performance of the underlying LLM components. Our human evaluation yields a 94% agreement rate for the

decomposition stage and an 92% agreement rate for the attribution stage, confirming the robustness of the atomic flow construction.

## 7. Directed Information Compression via Min-Cut

In modeling the RAG system as a flow network, we naturally look to maximize flow (FORD & Fulkerson, 1956), and by dual extension, we look to find a minimum cut. Further inspired by recent work on information-theoretic context engineering (Huang, 2025), we propose **Directed Information Compression** as a primary application of the Atomic Information Flow (AIF) model.

**Definition 7.1** (Information Capacity and Cut). Let the RAG system be represented as a graph where tools $t_i \in T$ are nodes with finite information capacity. We define the capacity $c(t_i)$ of a tool node as its probability of contributing relevant atoms to the final response:

$$c(t_i) \propto \mathbb{P}(t_i \text{ is utilized} \mid q, T). \quad (1)$$

A context cut partition $(S, \bar{S})$ divides the tools into a retained set $S$ (the active context) and a masked set $\bar{S}$ (the compressed context). The capacity of this cut, representing the total information loss, is the sum of the capacities of the

*Table 2.* ALCE vs AIF Tool Attribution Correctness. Metrics defined in H

| DATA SET | METRIC | ALCE VANILLA GPT5-NANO (BASELINE) | AIF GPT5-NANO (OURS) |
|---|---|---|---|
| **HOTPOTQA** | | | |
| TRUE | PRECISION | 89.9% | 89.2% |
| | RECALL | 75.1% | 76.2% |
| | F1 | 81.9% | 82.2% |
| FALSE | PRECISION | 64.5% | 69.7% |
| | RECALL | 55.7% | 59.5% |
| | F1 | 59.8% | 64.2% |
| **MSMARCO** | | | |
| TRUE | PRECISION | 35.1% | 39.3% |
| | RECALL | 81.6% | 87.2% |
| | F1 | 49.1% | 54.2% |
| FALSE | PRECISION | 7.3% | 8.4% |
| | RECALL | 84.9% | 86.6% |
| | F1 | 13.5% | 15.3% |
| **MUSIQUE** | | | |
| TRUE | PRECISION | 79.5% | 86.5% |
| | RECALL | 64.0% | 68.5% |
| | F1 | 70.9% | 76.5% |
| FALSE | PRECISION | 53.6% | 58.7% |
| | RECALL | 43.5% | 46.7% |
| | F1 | 48.0% | 52.0% |
| **WIKI MULTIHOP QA** | | | |
| TRUE | PRECISION | 91.9% | 80.1% |
| | RECALL | 78.2% | 76.8% |
| | F1 | 84.5% | 78.4% |
| FALSE | PRECISION | 53.5% | 57.4% |
| | RECALL | 53.8% | 56.4% |
| | F1 | 53.7% | 56.9% |

masked tools:

$$\text{Cut}(S, \bar{S}) = \sum_{t_j \in \bar{S}} c(t_j). \tag{2}$$

**Definition 7.2** (Directed Information Compression Policy). The compression policy $\pi(q, T)$ maps a query $q$ and tool set $T$ to an optimal subset $T' \subseteq T$. We formalize this policy as the solution to the Minimum Cut problem: finding the partition that minimizes the information loss of the masked set $\bar{S}$. This is equivalent to maximizing the likelihood that the retained set $T'$ satisfies the query:

$$T' = \pi(q, T) = \arg\min_{\bar{S} \subset T} \sum_{t_j \in \bar{S}} c(t_j) \tag{3}$$

$$\equiv \arg\max_{T' \subseteq T} \mathbb{P}(T' \mid q, T). \tag{4}$$

By solving for the minimum cut, the policy identifies and masks tools unlikely to influence the information flow from the query to the response super-sink, thereby minimizing token usage while preserving the critical atomic information required for answer correctness.

To operationalize this theoretical framework, we approximate the optimal compression policy $\pi(q, T)$ using a supervised learning approach. We employ Gemma3-4B (Team et al., 2024; 2025) as our policy model, training it to predict the optimal partition $(S, \bar{S})$ based on historical flow signals.

We construct training examples using the training splits from the datasets referenced in Table 3. The ground truth labels for the policy are derived directly from the AIF analysis: a tool node $t$ is assigned to the retained set $S$ if and only if it actively contributes atomic information to the final response. Formally, we select all $t$ such that the Tool Contribution metric $TUP_t \neq 0$ (see Table 1). This selection criterion acts as a proxy for maximizing the conditional probability $\mathbb{P}(T' \mid q, T)$ by ensuring that all sources of utilized information are preserved.

At inference time, the learned policy predicts the optimal subset $T'$ for unseen queries in the validation sets. We then enforce the computed cut by physically ablating the tools in the masked set $\bar{S} = T \setminus T'$ from the context window.

We use GPT4.1 as our answer generation model in these experiments, taking in the query $q$, and the masked set

$\bar{S} = T \setminus T'$ as the input context. The resulting performance trade-offs between token reduction and answer accuracy are reported in Table 3.

For clearer visualization, we showcase the trade-off between context compression and accuracy against larger, text-specialized compressor models in Figure 1, demonstrating the efficacy of the minimum-cut model as a training signal to bridge the gap exposed in smaller architectures.

| DATASET | TOKEN REDUCTION | ACCURACY |
|---|---|---|
| **HOTPOTQA** | | |
| FULL CONTEXT | 0% | 91.63% |
| GEMMA4B | 85.67% | 54.7% |
| GEMMA4B-AIF | 87.52% | 82.71% |
| **MS MARCOV2** | | |
| FULL CONTEXT | 0% | 69.46% |
| GEMMA4B | 51.9% | 49.8% |
| GEMMA4B-AIF | 66.31% | 50.55% |
| **MUSIQUE** | | |
| FULL CONTEXT | 0% | 70.74% |
| GEMMA4B | 87.7% | 16.9% |
| GEMMA4B-AIF | 90.72% | 36.79% |
| **WIKI QA** | | |
| FULL CONTEXT | 0% | 89.09% |
| GEMMA4B | 74.3% | 44.6% |
| GEMMA4B-AIF | 65.83% | 77.90% |

*Table 3.* **Minimum Cut Compression Results** - Token reduction and accuracy metrics for different compressor models. "Full Context" represents the upper bound (full context sent to answer LLM). All final answers are generated by feeding the compressor model's selected tool outputs to the GPT-4.1 model as **grounding_context** in Prompt E. **Accuracy** is computed against the final answer with LLM judge prompt as in Prompt F. **Token reduction** is the compression rate of the input context.

## 8. Limitations and Future work

### 8.1. Future Extension: Retrieval and Tool Calling

While our theoretical model in Figure 2 fully captures the flow from the user query to the final response, our experimental validation and post-training efforts in this paper focuses strictly on the Tool-to-Response edges (the "Generation" component). We leave the implementation and experimental validation of the Query-to-Tool edges (the "Retrieval" component) towards future work. Modeling the flow from $s_0 \to V_{\text{tool}}$ would allow for "Why did you retrieve this?" diagnostics, effectively extending AIF towards explainability in the tool calling and orchestration trajectories components of a RAG system. For this initial study, we prioritize the hallucination problem (verifying the answer against retrieved context) and only briefly touch upon the retrieval problem (verifying context against the query) G.

### 8.2. Fine Tuning Decomposition and Attribution Steps

Our current implementation leverages multiple LLM calls due to hidden output-length constraints on LLMs; however, the underlying tasks (atom segmentation, matching, labeling) are structurally simpler than general purpose text generation, suggesting a viable path toward fine-tuned lightweight models that can help reduce decomposition cost. We believe this to be a promising future direction to improve the scalability of the AIF pipeline.

### 8.3. Richer atomic signals

While we briefly explored relevance as a proof of concept in Appendix G, additional metadata signals—such as authority, recency, or uncertainty—could enable new flow heuristics and improve predictive correlations with answer correctness.

### 8.4. Trajectory Level Rewards

We eventually hope to develop trajectory level RL methods on top of AIF to allow for deeper explorations on high quality answers (Shao et al., 2024). One potential direction is to improve tool comprehensiveness in the answer generation phase by setting $RAR_K$ (1) as the reward function to compute advantages over.

## 9. Conclusion

In this work, we introduced **Atomic Information Flow (AIF)**, a network-flow framework for attributing information in retrieval-augmented and tool-augmented LLM systems at the level of minimal semantic units. Our experiments show that atomic flow metrics capture meaningful tool usage signals and expose attribution patterns that are not visible under conventional document-level grounding metrics, which are able to act as key signals to optimize the directed information flow between tool-response edges through post-training. We hope that AIF can set a theoretical model for trajectory level rewards, allowing for deeper opportunities for systematic optimizations throughout the entire RAG stack.

## Impact Statement

We believe that the atomic information flow framework and associated measurement techniques outlined in this paper have the potential to substantially shift the way researchers and practitioners conceptualize and develop modern tool based LLM systems. By introducing information flow at the atomic level and providing systematic, quantifiable mechanisms for attribution, contribution analysis, and optimization opportunities, our approach introduces a new paradigm for improving interpretability of complex language agent systems.

Beyond simple interpretability, we demonstrate some initial applications for atomic information flow, motivated by the minimum cut problem for context compression, demonstrating comparative compressing performance compared to models multiple times larger, implying better optimization opportunities for energy savings for sustainable development of the language model frontier.

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

## A. Atom Decomposition Prompt

```
# IMPORTANT - DO NOT STOP UNTIL EVERY SINGLE INFORMATION IS EXTRACTED

You are a grounding fact extractor and relevance score assigner. Your job is to:
1. Extract every informational statement from the provided grounding data,
    regardless of whether it's verifiable or not.
 - Extract every sentence in the grounding data as a factual statement, including
    preference and subjective information.
   - Each statement should be self-contained and understandable without additional
    context.
   - Replace all pronouns (he, she, it, they, his, her, its, their, etc.) and
    demonstrative references (this, that, these, those, the [noun]) with their
    corresponding subject entities so that each fact can stand alone without
    context (e.g., replace "the film" with the actual film name).
   - Preserve special terminology, links, and phrases in the extracted facts.
   - Individual words or phrases (such as meaningful titles or section names) must
    be understood within the context of the entire grounding data and form a
    coherent statement. Non-meaningful words or phrases should not be extracted as
    statements.
   - Do Not use any information from the user query when extracting facts.
   **Granularity Rules (apply in order):**
   a) **Keep together as single statements:**
      - Facts with causal relationships (e.g., "Because of X, Y occurred")
      - Facts with temporal relationships (e.g., "When X began, Y happened")
      - Facts with conditional relationships (e.g., "If X, then Y")
      - Multiple items that share the same relationship to a subject (e.g., "John
    played in Musical A, Musical B, and Musical C")
   b) **Separate into distinct statements:**
      - Items that have different relationships to a subject (e.g., "John played
    in Musical A" and "John directed Musical B" should be two statements)
      - Items that have different contexts or timeframes
      - Facts that can stand independently without losing meaning
   c) **When in doubt:**
      - Prioritize self-containment: the statement must make complete sense on its
    own
      - Prioritize preserving logical relationships: don't break apart facts that
    depend on each other for meaning

2. For each factual statement, assign a relevance score to the user query using
    the following Likert scale:
   - 1: Not relevant at all (the fact has no connection to the user query)
   - 2: Slightly relevant (the fact is tangentially related but not useful for
    answering the query)
   - 3: Moderately relevant (the fact is somewhat related and may provide partial
    context)
   - 4: Very relevant (the fact directly supports or answers part of the query)
   - 5: Extremely relevant (the fact is essential and directly answers the query)

### Output Format
[
    {{
        Fact: Natural Language Fact from grounding data,
        Relevance: 3
    }}
]
Output the Json array only, without any additional text or explanation.

### Examples

#### Example 1
```

```
User Query: Why is my video streaming account suspended?
Grounding Data: The StreamView account is suspended due to a failed payment
    transaction. Access is restored immediately once a valid card is added.
    Account suspension does not cancel the billing cycle. You can update payment
    details in the settings tab. Pending charges are re-attempted every 48 hours.
    Support cannot manually override suspension without payment. It is recommended
    to check with your bank for blocks.
Output:
[
  {{ "Fact": "The StreamView account is suspended due to a failed payment
    transaction.", "Relevance": 5 }},
  {{ "Fact": "Access is restored immediately once a valid card is added.",
    "Relevance": 5 }},
  {{ "Fact": "Account suspension does not cancel the billing cycle.", "Relevance":
    4 }},
  {{ "Fact": "You can update payment details in the settings tab.", "Relevance": 4
    }},
  {{ "Fact": "Pending charges are re-attempted every 48 hours.", "Relevance": 3 }},
  {{ "Fact": "Support cannot manually override suspension without payment.",
    "Relevance": 3 }},
  {{ "Fact": "It is recommended to check with your bank for blocks.", "Relevance":
    2 }}
]

#### Example 2
User Query: What features are included in the new Smart Home Hub?
Grounding Data: The Smart Home Hub features voice control and energy monitoring.
    The launch announcement mentions compatibility with third-party bulbs. A
    mobile app is required for setup. The Hub connects via Wi-Fi and Bluetooth.
    Security cameras and door locks can be integrated. Customer reviews highlight
    ease of use. The device comes in white and black colors. Standard warranty
    covers one year.
Output:
[
  {{ "Fact": "The Smart Home Hub features voice control and energy monitoring.",
    "Relevance": 5 }},
  {{ "Fact": "The Smart Home Hub connects via Wi-Fi and Bluetooth.", "Relevance":
    5 }},
  {{ "Fact": "A mobile app is required for setup.", "Relevance": 4 }},
  {{ "Fact": "The launch announcement mentions compatibility with third-party
    bulbs.", "Relevance": 4 }},
  {{ "Fact": "Security cameras and door locks can be integrated.", "Relevance": 4
    }},
  {{ "Fact": "Customer reviews highlight ease of use.", "Relevance": 3 }},
  {{ "Fact": "The device comes in white and black colors.", "Relevance": 2 }},
  {{ "Fact": "Standard warranty covers one year.", "Relevance": 2 }}
]

#### Example 3
User Query: Tell me about the Mars Perseverance Rover
Grounding Data: Perseverance is a Mars rover developed by NASA. It landed on Mars
    in February 2021. Its mission is to seek signs of ancient life. It carries a
    small helicopter named Ingenuity. The rover is about the size of a car.
Output:
[
  {{ "Fact": "Perseverance is a Mars rover developed by NASA.", "Relevance": 5 }},
  {{ "Fact": "Perseverance landed on Mars in February 2021.", "Relevance": 4 }},
  {{ "Fact": "Perseverance's mission is to seek signs of ancient life.",
    "Relevance": 5 }},
  {{ "Fact": "Perseverance carries a small helicopter named Ingenuity.",
    "Relevance": 4 }},
  {{ "Fact": "The rover is about the size of a car.", "Relevance": 3 }}
```

```
770    ]
771
772
773    #### Example 4
774    User Query: How do I recover a deleted file?
775    Grounding Data: The system has a recycle bin that holds files for 30 days. You can
776        restore files by right-clicking them in the bin. Permanently deleted files
777        cannot be recovered without specialized software. Cloud backups may contain
778        older versions. The interface supports custom themes.
779    Output:
780    [
781      {{ "Fact": "The system has a recycle bin that holds files for 30 days.",
782        "Relevance": 5 }},
783      {{ "Fact": "You can restore files by right-clicking them in the bin.",
784        "Relevance": 5 }},
785      {{ "Fact": "Permanently deleted files cannot be recovered without specialized
786        software.", "Relevance": 4 }},
787      {{ "Fact": "Cloud backups may contain older versions.", "Relevance": 4 }},
788      {{ "Fact": "The interface supports custom themes.", "Relevance": 1 }}
789    ]
790
791    ### User Query
792    {user_query}
793
794    ### Grounding Data
795    {grounding_data}
```

## B. Response Atom Assignment Prompt

```
You are a fact extractor and fact assignment checker for LLM evaluation. Given a
    **response** to a **query** and a list of **grounding facts**, your task is to:
1. Break down the **response** into discrete fact/claim statements.
   - Extract and output only the main findings and results, excluding any initial
     planning, task breakdown, assumptions, or next steps.
   - Each statement should be self-contained and understandable without additional
     context.
   - Replace all pronouns (he, she, it, they, his, her, its, their, etc.) and
     demonstrative references (this, that, these, those, the [noun]) with their
     corresponding subject entities.
   - Preserve special terminology, links, and phrases in the extracted statement.
   - Individual words or phrases (such as meaningful titles or section names) must
     be understood within the context of the entire response and form a coherent
     statement. Non-meaningful words or phrases should not be extracted as
     statements.
   - Don't stop generation until every single piece of information is extracted.
   **Granularity Rules (apply in order):**
   a) **Keep together as single statements:**
      - Facts with causal relationships (e.g., "Because of X, Y occurred")
      - Facts with temporal relationships (e.g., "When X began, Y happened")
      - Facts with conditional relationships (e.g., "If X, then Y")
      - Multiple items that share the same relationship to a subject (e.g., "John
      played in Musical A, Musical B, and Musical C")
   b) **Separate into distinct statements:**
      - Items that have different relationships to a subject (e.g., "John played in
      Musical A" and "John directed Musical B" should be two statements)
      - Items that have different contexts or timeframes
      - Facts that can stand independently without losing meaning
   c) **When in doubt:**
      - Prioritize self-containment: the statement must make complete sense on its
      own
```

```
       – Prioritize preserving logical relationships: don't break apart facts that
       depend on each other for meaning
2. For each statement in the response, identify ALL grounding facts that support
   it (fully or partially).
   Assignment Process:
   - Use the 0-based index provided in front of each grounding fact (e.g., "Index
     0:", "Index 1:")
   - Return a list of ALL supporting fact indices (e.g., [0, 2, 5])
   - Return an empty list [] if no grounding facts support the statement
   - A fact "supports" a statement if it provides evidence for any part of that
     statement
   Inference and Reasoning:
   - A statement can be supported through DIRECT matching OR through INFERENCE from
     multiple facts
   - Direct support: The grounding fact explicitly states the information (e.g.,
     "User is in Netherlands"  Index: "User is based in Netherlands")
   - Inference support: The statement can be logically derived by combining
     multiple grounding facts
      Example: Statement "User joined in 2022" can be inferred from Index 0: "User
     has 3 years tenure" + Index 1: "Current year is 2025"
      Example: Statement "User prefers dark mode UI" can be inferred from Index 2:
     "User always disables light themes" + Index 3: "User customizes to dark colors"
      Example: Statement "Director A was born later than Director B" can be
     inferred from Index 0: "Director A born 1946" + Index 1: "Director B born
     1910" (comparison: 1946 > 1910)
   - Include ALL facts used in the reasoning chain, even if individually they don't
     fully support the statement
3. Output your assignments as a JSON array with the following structure:
### Output Format
[
    {{
        "Fact": "Verbatim statement from the response",
        "Assignment": [0, 2, 5]  // List of supporting fact indices, or []
    }}
]
Field Definitions:
* "Fact": The exact factual statement from the response (word-for-word)
* "Assignment": List of 0-based grounding fact indices that support this statement
    (directly or through inference), or [] if none
Output ONLY the JSON array, without any additional text or explanation.

### Examples

#### Example 1: Basic Fact Assignment
Query:
"Who founded SpaceX and when?"

Response:
"SpaceX was founded in 2002 by Elon Musk. He enjoys playing video games."

Grounding Facts:
[
  "Index 0: SpaceX was founded by Elon Musk",
  "Index 1: SpaceX was established in the year 2002",
  "Index 2: Elon Musk is the CEO of Tesla"
]

Output:
[
  {{"Fact": "SpaceX was founded by Elon Musk", "Assignment": [0]}},
  {{"Fact": "SpaceX was founded in 2002", "Assignment": [1]}},
  {{"Fact": "Elon Musk founded SpaceX in 2002", "Assignment": [0, 1]}},
```

```
    {{"Fact": "Elon Musk enjoys playing video games", "Assignment": []}}
]

Explanation:
- Fact 1: Directly matches the founder claim in Index 0
- Fact 2: Directly matches the year claim in Index 1
- Fact 3: Inferred by combining Index 0 (founder) + Index 1 (year) to validate the
    full statement
- Fact 4: No grounding fact mentions video games (cannot be inferred from context)

#### Example 2: Complex Inference with Tenure Calculation
Query:
"How long has Satya Nadella led Microsoft and what is his background?"

Response:
"Satya Nadella has led Microsoft for over 10 years and previously worked on cloud
    infrastructure."

Grounding Facts:
[
  "Index 0: Satya Nadella was named CEO of Microsoft in February 2014.",
  "Index 1: The current date is March 2024.",
  "Index 2: Before becoming CEO, Nadella was Executive Vice President of
   Microsoft's Cloud and Enterprise group.",
  "Index 3: Nadella originally joined Microsoft in 1992.",
  "Index 4: Microsoft is headquartered in Redmond, Washington."
]

Output:
[
  {{"Fact": "Satya Nadella has led Microsoft for over 10 years", "Assignment": [0,
   1]}},
  {{"Fact": "Satya Nadella previously worked on cloud infrastructure",
   "Assignment": [2]}}
]

Explanation:
- Fact 1: Inferred by calculating tenure from Index 0 (CEO start Feb 2014) + Index
    1 (current date Mar 2024) = ~10 years and 1 month.
- Fact 2: Inferred from Index 2 (EVP of Cloud group = worked on cloud
    infrastructure).

#### Example 3: Multi-Fact Inference with Birth Date Comparison
Query:
"Which film has the director who was born later, Best Man Wins or Mrs Caldicot's
    Cabbage War?"

Response:
"The film with the director born later is Mrs Caldicot's Cabbage War. Its
    director, Ian Sharp, was born in 1946, while Best Man Wins was directed by
    John Sturges, who was born in 1910."

Grounding Facts:
[
  "Index 0: Best Man Wins is a 1948 American historical drama film directed by
   John Sturges, based on a story by Mark Twain.",
  "Index 1: Ian Sharp was born on 13 November 1946 in Clitheroe, Lancashire.",
  "Index 2: Ian Sharp is an English film and television director.",
  "Index 3: John Eliot Sturges was born on January 3, 1910.",
  "Index 4: John Eliot Sturges was an American film director.",
  "Index 5: Mrs Caldicot's Cabbage War is a British comedy-drama film from 2002
   directed by Ian Sharp and starring Pauline Collins, John Alderton and Peter
```

```
      Capaldi."
    ]

    Output:
    [
      {{"Fact": "The film with the director born later is Mrs Caldicot's Cabbage War",
        "Assignment": [1, 3, 5]}},
      {{"Fact": "Mrs Caldicot's Cabbage War's director, Ian Sharp, was born in 1946",
        "Assignment": [1, 5]}},
      {{"Fact": "Best Man Wins was directed by John Sturges", "Assignment": [0, 4]}},
      {{"Fact": "John Sturges was born in 1910", "Assignment": [3]}}
    ]

    Explanation:
    - Fact 1: Inferred by comparing birth years from Index 1 (Ian Sharp born 1946) and
        Index 3 (John Sturges born 1910), combined with Index 5 (Ian Sharp directed
        Mrs Caldicot's Cabbage War). The comparison "born later" (1946 > 1910) is
        derived from these facts.
    - Fact 2: Directly combines Index 1 (Ian Sharp's birth year) + Index 5 (Ian Sharp
        directed Mrs Caldicot's Cabbage War)
    - Fact 3: Combines Index 0 (Best Man Wins directed by John Sturges) + Index 4
        (John Sturges is a director)
    - Fact 4: Directly matches Index 3 (John Eliot Sturges was born on January 3, 1910)

    #### Example 4: Multi-Fact Inference with Death Date Comparison
    Query:
    "Which director died first, the director of Lucky Jordan or the director of
        Adventures of Joselito and Tom Thumb?"

    Response:
    "The director of \"Lucky Jordan,\" Frank Tuttle, died first. Frank Tuttle died on
        January 6, 1963, while Ren Cardona, the director of \"Adventures of Joselito
        and Tom Thumb,\" died on April 25, 1988."

    Grounding Facts:
    [
      "Index 0: Lucky Jordan is a 1942 film directed by Frank Tuttle.",
      "Index 1: Lucky Jordan stars Alan Ladd in his first leading role.",
      "Index 2: Frank Wright Tuttle was a Hollywood film director and writer.",
      "Index 3: Frank Wright Tuttle was born on August 6, 1892.",
      "Index 4: Frank Wright Tuttle died on January 6, 1963.",
      "Index 5: Ren Cardona was born on October 8, 1905 in Havana, Cuba.",
      "Index 6: Ren Cardona died on April 25, 1988 in Mexico City.",
      "Index 7: Ren Cardona was a director in the Golden Age of Mexican cinema.",
      "Index 8: Adventures of Joselito and Tom Thumb is a 1960 Mexican-Spanish musical
        film directed by Ren Cardona and starring Joselito, Cesreo Quezadas and
        Enrique Rambal.",
      "Index 9: Tomy's Secret is a 1963 French-Spanish musical film directed by
        Antonio del Amo and starring Fabienne Dali, Joselito and Fernando Casanova."
    ]

    Output:
    [
      {{"Fact": "The director of \"Lucky Jordan,\" Frank Tuttle, died first",
        "Assignment": [0, 4, 6, 8]}},
      {{"Fact": "Frank Tuttle died on January 6, 1963", "Assignment": [4]}},
      {{"Fact": "Ren Cardona, the director of \"Adventures of Joselito and Tom
        Thumb,\" died on April 25, 1988", "Assignment": [6, 8]}}
    ]

    Explanation:
    - Fact 1: Inferred by comparing death dates from Index 4 (Frank Tuttle died 1963)
```

```
      and Index 6 (Ren Cardona died 1988), combined with Index 0 (Frank Tuttle
      directed Lucky Jordan) and Index 8 (Ren Cardona directed Adventures of
      Joselito and Tom Thumb). The comparison "died first" (1963 < 1988) is derived
      from these facts.
    – Fact 2: Directly matches Index 4 (Frank Wright Tuttle died on January 6, 1963)
    – Fact 3: Directly combines Index 6 (Ren Cardona's death date) + Index 8 (Ren
      Cardona directed Adventures of Joselito and Tom Thumb)

    ### Query
    {query}
    ### Response
    {response}
    ### Grounding Fact List
    {grounding_facts}
```

## C. Example Tools from HotPotQA

```
[
  {
    "tool_name": "Loretta Lynn",
    "tool_input": "{query:  Here I Am Again is an album by what American country
    singer who had a career of almost 60 years}",
    "tool_output": "Loretta Lynn (ne Webb; born April 14, 1932) is an American
    country music singer-songwriter with multiple gold albums over a career of
    almost 60 years. She has received numerous awards and other accolades for her
    groundbreaking role in country music, including awards from both the Country
    Music Association and Academy of Country Music as a duet partner and an
    individual artist. She is the most awarded female country recording artist and
    the only female ACM Artist of the Decade (1970s).",
    "tool_metadata": "",
    "tool_execution_duration": 0
  },
  {
    "tool_name": "Arthur Jensen (actor)",
    "tool_input": "{query:  Here I Am Again is an album by what American country
    singer who had a career of almost 60 years}",
    "tool_output": "Arthur Jensen (9 November 1897  28 November 1981) was a Danish
    actor whose career lasted for almost 60 years. He made his dbut on stage at
    the Royal Danish Theatre in 1923, and he had his big screen dbut in the silent
    film \"Pas p pigerne\" in 1930.",
    "tool_metadata": "",
    "tool_execution_duration": 0
  },
  {
    "tool_name": "Here I Am Again",
    "tool_input": "{query:  Here I Am Again is an album by what American country
    singer who had a career of almost 60 years}",
    "tool_output": "Here I Am Again is the twentieth studio album by American
    country music singer-songwriter, Loretta Lynn. It was released on October 2,
    1972, by Decca Records. This would be Lynn's last studio album with Decca
    Records, which would merge with MCA Records in 1973.",
    "tool_metadata": "",
    "tool_execution_duration": 0
  },
  {
    "tool_name": "Desmond Elliott",
    "tool_input": "{query:  Here I Am Again is an album by what American country
    singer who had a career of almost 60 years}",
    "tool_output": "Desmond Elliott (1930  2003) was a distinguished publisher and
```

```
      literary agent. Having started his career at the publishing house Macmillan,
      he later went on to found his own publishing company, Arlington Books. In a
      career of over almost 60 years he was responsible for discovering a number of
      writers who went on to be bestsellers, including Penny Vincenzi and Jilly
      Cooper.",
       "tool_metadata": "",
       "tool_execution_duration": 0
     },
     {
       "tool_name": "Brenda Lee",
       "tool_input": "{query:  Here I Am Again is an album by what American country
       singer who had a career of almost 60 years}",
       "tool_output": "Brenda Lee (born Brenda Mae Tarpley, December 11, 1944) is an
       American performer and the top-charting solo female vocalist of the 1960s. She
       sang rockabilly, pop and country music, and had 47 US chart hits during the
       1960s, and is ranked fourth in that decade surpassed only by Elvis Presley,
       the Beatles and Ray Charles. She is perhaps best known in the United States
       for her 1960 hit \"I'm Sorry\", and 1958's \"Rockin' Around the Christmas
       Tree\", a United States holiday standard for almost 60 years.",
       "tool_metadata": "",
       "tool_execution_duration": 0
     },
     {
       "tool_name": "Sergey Mikhalkov",
       "tool_input": "{query:  Here I Am Again is an album by what American country
       singer who had a career of almost 60 years}",
       "tool_output": "Sergey Vladimirovich Mikhalkov (Russian:   ; 13 March [O.S. 28
       February] 1913  27 August 2009) was a Soviet and Russian author of children's
       books and satirical fables who had the opportunity to write the lyrics of his
       country's national anthem on three different occasions, spanning almost 60
       years.",
       "tool_metadata": "",
       "tool_execution_duration": 0
     },
     {
       "tool_name": "Philip Jos Farmer bibliography",
       "tool_input": "{query:  Here I Am Again is an album by what American country
       singer who had a career of almost 60 years}",
       "tool_output": "In a writing career spanning more than 60 years (19462008),
       American science fiction and fantasy author Philip Jos Farmer published almost
       60 novels, over 100 short stories and novellas (many expanded or combined into
       novels), two \"fictional biographies\", and numerous essays, articles and
       ephemera in fan publications.",
       "tool_metadata": "",
       "tool_execution_duration": 0
     },
     {
       "tool_name": "Alan Whicker",
       "tool_input": "{query:  Here I Am Again is an album by what American country
       singer who had a career of almost 60 years}",
       "tool_output": "Alan Donald Whicker (2 August 1921  12 July 2013) was a
       British journalist and television presenter and broadcaster. His career
       spanned almost 60 years, during which time he presented the documentary
       television programme \"Whicker's World\" for over 30 years. He was made a
       Commander of the Order of the British Empire (CBE) in 2005 for services to
       broadcasting.",
       "tool_metadata": "",
       "tool_execution_duration": 0
     },
     {
       "tool_name": "Robert Sutton (Irish judge)",
       "tool_input": "{query:  Here I Am Again is an album by what American country
```

```
      singer who had a career of almost 60 years}",
       "tool_output": "Robert Sutton (c. 1340  1430) was an Irish judge and Crown
      official. During a career which lasted almost 60 years he served the Crown in
      a variety of offices, notably as Deputy to the Lord Chancellor of Ireland,
      Chief Baron of the Irish Exchequer, Master of the Rolls in Ireland, and Deputy
      Treasurer of Ireland. A royal warrant of 1423 praises his \"long and
      laudable\" service to the Crown.",
       "tool_metadata": "",
       "tool_execution_duration": 0
    },
    {
       "tool_name": "Pete Alvarado",
       "tool_input": "{query:  Here I Am Again is an album by what American country
      singer who had a career of almost 60 years}",
       "tool_output": "Peter J. Alvarado, Jr. (February 22, 1920  December 27, 2003)
      was an American animation and comic book artist. Alvarado's animation career
      spanned almost 60 years. He was also a prolific contributor to Western
      Publishing's line of comic books.",
       "tool_metadata": "",
       "tool_execution_duration": 0
    }
  ]
```

## D. Full Tool Decomposition of C

```
  [
    {
       "Index": 0,
       "Fact": "Arthur Jensen (9 November 1897  28 November 1981) was a Danish actor
      whose career lasted for almost 60 years.",
       "Relevance": 1
    },
    {
       "Index": 1,
       "Fact": "Arthur Jensen made his dbut on stage at the Royal Danish Theatre in
      1923.",
       "Relevance": 1
    },
    {
       "Index": 2,
       "Fact": "Arthur Jensen had his big screen dbut in the silent film Pas p
      pigerne in 1930.",
       "Relevance": 1
    },
    {
       "Index": 3,
       "Fact": "Loretta Lynn (ne Webb; born April 14, 1932) is an American country
      music singer-songwriter with multiple gold albums over a career of almost 60
      years.",
       "Relevance": 5
    },
    {
       "Index": 4,
       "Fact": "Loretta Lynn has received numerous awards and other accolades for her
      groundbreaking role in country music, including awards from both the Country
      Music Association and Academy of Country Music as a duet partner and an
      individual artist.",
       "Relevance": 4
    },
```

```
      {
        "Index": 5,
        "Fact": "Loretta Lynn is the most awarded female country recording artist and
        the only female ACM Artist of the Decade (1970s).",
        "Relevance": 4
      },
      {
        "Index": 6,
        "Fact": "Peter J. Alvarado, Jr. (February 22, 1920  December 27, 2003) was an
        American animation and comic book artist.",
        "Relevance": 1
      },
      {
        "Index": 7,
        "Fact": "Peter J. Alvarado, Jr.'s animation career spanned almost 60 years.",
        "Relevance": 2
      },
      {
        "Index": 8,
        "Fact": "Peter J. Alvarado, Jr. was also a prolific contributor to Western
        Publishing's line of comic books.",
        "Relevance": 2
      },
      {
        "Index": 9,
        "Fact": "Alan Donald Whicker (2 August 1921  12 July 2013) was a British
        journalist and television presenter and broadcaster.",
        "Relevance": 1
      },
      {
        "Index": 10,
        "Fact": "Alan Whicker's career spanned almost 60 years, during which he
        presented the documentary television programme 'Whicker's World' for over 30
        years.",
        "Relevance": 1
      },
      {
        "Index": 11,
        "Fact": "Alan Whicker was made a Commander of the Order of the British Empire
        (CBE) in 2005 for services to broadcasting.",
        "Relevance": 1
      },
      {
        "Index": 12,
        "Fact": "Here I Am Again is the twentieth studio album by American country
        music singer-songwriter, Loretta Lynn.",
        "Relevance": 5
      },
      {
        "Index": 13,
        "Fact": "Here I Am Again was released on October 2, 1972, by Decca Records.",
        "Relevance": 3
      },
      {
        "Index": 14,
        "Fact": "Here I Am Again would be Loretta Lynn's last studio album with Decca
        Records.",
        "Relevance": 5
      },
      {
        "Index": 15,
        "Fact": "Decca Records would merge with MCA Records in 1973.",
```

```
      "Relevance": 2
    },
    {
      "Index": 16,
      "Fact": "Robert Sutton lived circa 1340 to 1430.",
      "Relevance": 1
    },
    {
      "Index": 17,
      "Fact": "Robert Sutton was an Irish judge and Crown official.",
      "Relevance": 1
    },
    {
      "Index": 18,
      "Fact": "Robert Sutton's career lasted almost 60 years.",
      "Relevance": 2
    },
    {
      "Index": 19,
      "Fact": "Robert Sutton served the Crown in a variety of offices, notably as
     Deputy to the Lord Chancellor of Ireland, Chief Baron of the Irish Exchequer,
     Master of the Rolls in Ireland, and Deputy Treasurer of Ireland.",
      "Relevance": 2
    },
    {
      "Index": 20,
      "Fact": "A royal warrant of 1423 praises Robert Sutton's 'long and laudable'
     service to the Crown.",
      "Relevance": 2
    },
    {
      "Index": 21,
      "Fact": "Philip Jos Farmer is an American science fiction and fantasy author.",
      "Relevance": 1
    },
    {
      "Index": 22,
      "Fact": "Philip Jos Farmer's writing career spanned more than 60 years
     (19462008).",
      "Relevance": 2
    },
    {
      "Index": 23,
      "Fact": "Philip Jos Farmer published almost 60 novels, over 100 short stories
     and novellas (many expanded or combined into novels), two 'fictional
     biographies', and numerous essays, articles and ephemera in fan publications.",
      "Relevance": 2
    },
    {
      "Index": 24,
      "Fact": "Brenda Lee's birth name is Brenda Mae Tarpley.",
      "Relevance": 2
    },
    {
      "Index": 25,
      "Fact": "Brenda Lee was born on December 11, 1944.",
      "Relevance": 2
    },
    {
      "Index": 26,
      "Fact": "Brenda Lee is an American performer.",
      "Relevance": 5
```

```
        },
        {
          "Index": 27,
          "Fact": "Brenda Lee is the top-charting solo female vocalist of the 1960s.",
          "Relevance": 3
        },
        {
          "Index": 28,
          "Fact": "Brenda Lee sang rockabilly, pop, and country music.",
          "Relevance": 5
        },
        {
          "Index": 29,
          "Fact": "Brenda Lee had 47 US chart hits during the 1960s.",
          "Relevance": 4
        },
        {
          "Index": 30,
          "Fact": "Brenda Lee is ranked fourth in the 1960s, surpassed only by Elvis
         Presley, the Beatles and Ray Charles.",
          "Relevance": 3
        },
        {
          "Index": 31,
          "Fact": "Brenda Lee is best known in the United States for her 1960 hit \"I'm
         Sorry\".",
          "Relevance": 2
        },
        {
          "Index": 32,
          "Fact": "Brenda Lee's 1958 song \"Rockin' Around the Christmas Tree\" is a
         United States holiday standard.",
          "Relevance": 3
        },
        {
          "Index": 33,
          "Fact": "Brenda Lee's 1958 song \"Rockin' Around the Christmas Tree\" has been
         a United States holiday standard for almost 60 years.",
          "Relevance": 4
        },
        {
          "Index": 34,
          "Fact": "Sergey Vladimirovich Mikhalkov was a Soviet and Russian author of
         children's books and satirical fables.",
          "Relevance": 1
        },
        {
          "Index": 35,
          "Fact": "Sergey Vladimirovich Mikhalkov had the opportunity to write the
         lyrics of his country's national anthem on three different occasions spanning
         almost 60 years.",
          "Relevance": 1
        },
        {
          "Index": 36,
          "Fact": "Sergey Vladimirovich Mikhalkov's name in Russian is   .",
          "Relevance": 1
        },
        {
          "Index": 37,
          "Fact": "Sergey Vladimirovich Mikhalkov was born on 13 March [O.S. 28
         February] 1913.",
```

```
      "Relevance": 1
    },
    {
      "Index": 38,
      "Fact": "Sergey Vladimirovich Mikhalkov died on 27 August 2009.",
      "Relevance": 1
    },
    {
      "Index": 39,
      "Fact": "Desmond Elliott (1930  2003) was a distinguished publisher and
      literary agent.",
      "Relevance": 1
    },
    {
      "Index": 40,
      "Fact": "Desmond Elliott started his career at the publishing house
      Macmillan.",
      "Relevance": 1
    },
    {
      "Index": 41,
      "Fact": "Desmond Elliott later founded his own publishing company, Arlington
      Books.",
      "Relevance": 1
    },
    {
      "Index": 42,
      "Fact": "Desmond Elliott had a career of almost 60 years during which he was
      responsible for discovering a number of writers who went on to be bestsellers,
      including Penny Vincenzi and Jilly Cooper.",
      "Relevance": 1
    }
  ]
```

## E. QA Answer Generation Prompt

```
You are a helpful assistant that provides accurate, concise answers based on the
    provided grounding context.

Your task is to answer the user's query using only the information provided in the
    grounding context.

**Important Guidelines:**
- Respond in full sentences. Do not use bullet points or lists.
- Only use information that is directly supported by the grounding context
- Answer to the best of your ability, do not hallucinate or make up information.

### User Query
{user_query}

### Grounding Context
{grounding_context}
```

# F. LLM Judge Prompt

```
You are an answer correctness evaluator. Your task is to determine if an actual
    answer is correct with respect to a reference answer for a given query.

**Evaluation Criteria:**
- The actual answer should be semantically equivalent to the reference answer,
    even if the wording differs
- The actual answer should correctly address the query
- Minor differences in phrasing or formatting should not affect correctness
- The actual answer must contain the same key information as the reference answer

**Output Format:**
You must output ONLY a boolean value: `true` or `false`
- `true` if the actual answer is correct with respect to the reference answer
- `false` if the actual answer is incorrect, incomplete, or does not match the
    reference answer
```

# G. Relevance Labeling

We initially hypothesized that LLM assigned relevance labels would carry some predictive power on answer correctness by chaining atomic attribution from tools with high relevance labels. Interestingly, we found that setting high K values gradually resulted in weaker predictive power on the True/False segments of the Musique dataset (Trivedi et al., 2022) - with $RAP@K = 0$ carrying the strongest predictive power as a measure of pure faithfulness to tools. That being said, we do believe there is still a strong opportunity to explore additional metadata features such as authority, recency, or other metadata to further enrich the atom representation to gain additional predictive power. The AUROC plot in Figure 5 shows the predictive power of the RAP@K metric on answer correctness. We theorize that future directions may find value in leveraging tool-specific atomic signals for generating additional flow heuristics.

# H. Experimental Metrics

Let $G_i$ denote the set of ground truth attributions for example $i$, and $P_i$ denote the set of predicted attributions for example $i$. Let $N$ denote the total number of examples. Then:

**Definition H.1** (Precision). **Precision** is defined as the average proportion of predicted attributions that are correct:

$$\text{Precision} = \frac{1}{N} \sum_{i=1}^{N} \frac{|G_i \cap P_i|}{|P_i|}$$

where $|G_i \cap P_i|$ is the number of correctly predicted attributions for example $i$, and $|P_i|$ is the total number of predicted attributions for $i$.

**Definition H.2** (Recall). **Recall** is defined as the average proportion of ground truth attributions that are correctly predicted:

$$\text{Recall} = \frac{1}{N} \sum_{i=1}^{N} \frac{|G_i \cap P_i|}{|G_i|}$$

where $|G_i \cap P_i|$ is the number of correctly predicted attributions for example $i$, and $|G_i|$ is the total number of ground truth attributions for $i$.

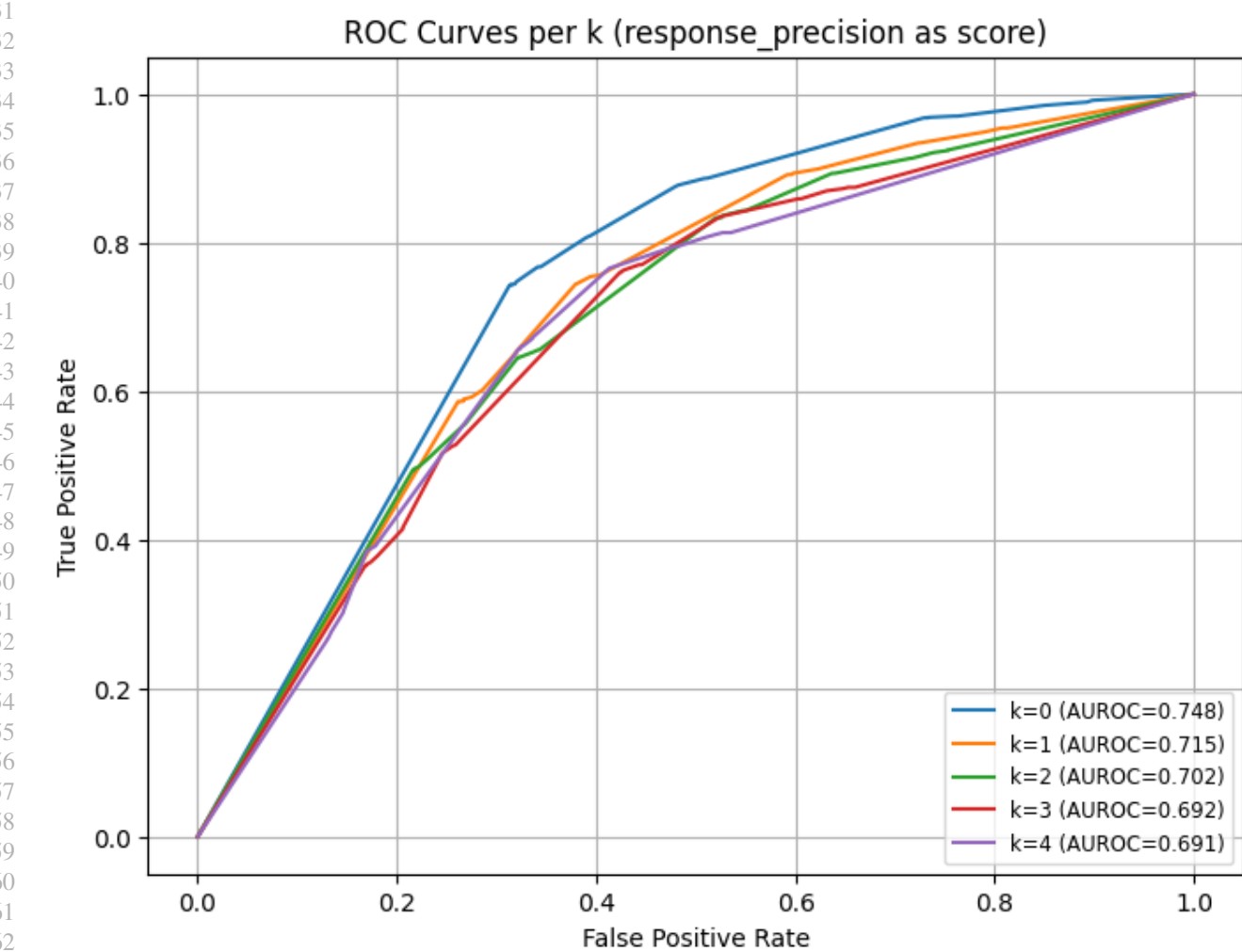

*Figure 5.* AUROC for LLM Assigned Relevance Labels Against Correct/Incorrect Responses on Musique Dataset

## I. Additional Insights from Directed Information Compression Results

We trained Gemma3-4B (Team et al., 2024; 2025) to predict the optimal tool subset $T'$ using four datasets: HotPotQA, MS MARCO, 2WikiMultihopQA, and MuSiQue. Since HotPotQA served as our pivot dataset, we tuned our fact attribution prompt toward applying more reasoning during attribution. As shown in Table 3:

**Multi-hop reasoning datasets** show substantial improvements after compressor model training:

- Accuracy gains: +19.89% to +33.30% (HotPotQA: 54.7% → 82.71%)

- Token reduction maintained: 65.83%-90.72%

**MS MARCO**, a retrieval-focused dataset without multi-hop reasoning requirements, exhibits marginal accuracy improvement (+0.75%) but substantial token reduction gain (+14.41%: 51.9% → 66.31%). This suggests that retrieval contexts contain more redundancy than reasoning contexts, allowing aggressive compression with minimal accuracy impact.

These results demonstrate that AIF-based minimum-cut signals provide effective training supervision for context compression, bridging the performance gap between small and large models while maintaining high compression rates.

