# OpenReview forum: "Atomic Information Flow: A Network Flow Model for Tool Attributions in RAG Systems"
_ICML.cc/2026/Conference — Submitted to ICML 2026_

### Official Review · Reviewer_aXiE · 2026-03-12

**Soundness:** 2
**Presentation:** 2
**Significance:** 2
**Originality:** 3
**Overall Recommendation:** 3
**Confidence:** 4

**Summary:**

The paper proposes an Atomic Information Flow framework to trace which parts of the response came from which tools in a tool-based RAG system. The paper adopts a graph-based network flow model, leveraging max-flow min-cut duality, to model information flow following three stages, including (1) atomic decomposition of tool output, (2) atomic signal injection, and (3) response atom assignment. Moreover, the paper also proposes directed information compression as an application of the Atomic Information Flow model. A thorough flow heuristic has been proposed along with four QA datasets for experiments, and shows experiments comparing to ALCE Vanilla GPT5-Nano and achieves better performance for most cases.

**Compliance With Llm Reviewing Policy:**

Affirmed.

**Final Justification:**

My concerns and questions raised during rebuttal are largely unaddressed, thus I would like to maintain my current score.

**Key Questions For Authors:**

- In Table 2, what is the overall improvement of your Method compared to the baseline? Can you provide intuition for why your method dropped 11.8% in precision on Wiki Multihop QA?

**Limitations:**

Yes

**Strengths And Weaknesses:**

Strength
- The theoretical foundations are carefully laid out. The formal definitions, including atoms, supply nodes, super-source, etc are precise and consistent.
- The paper is well structured, and the narrative, including motivation and formal models, is clear.
- The paper addresses an important and interesting problem.

Weakness
- The paper mentioned tool calls in the RAG system, while the experiments ignore the differences between various tools, focusing only on which source is useful for answering certain part of the final answer.
- The paper only compares with GPT5-Nano, and in the application section "Directed Information Compression via Min-Cut", no baselines are compared. Many approaches have been proposed to understand which information from the context is useful for answering the questions, specifically in the hard prompting domain such as:
[1] Jiang, Huiqiang, et al. "Llmlingua: Compressing prompts for accelerated inference of large language models." Proceedings of the 2023 conference on empirical methods in natural language processing. 2023.
[2] Xu, Fangyuan, Weijia Shi, and Eunsol Choi. "Recomp: Improving retrieval-augmented lms with compression and selective augmentation." arXiv preprint arXiv:2310.04408 (2023).
[3] Choi, Eunseong, et al. "From reading to compressing: Exploring the multi-document reader for prompt compression." Findings of the Association for Computational Linguistics: EMNLP 2024. 2024.
[4] Tang, Jiwei, et al. "Perception compressor: A training-free prompt compression framework in long context scenarios." Findings of the Association for Computational Linguistics: NAACL 2025. 2025.
- LLM and network pipe are essentially different when discussing in Definition 2.8. How do you model the information within each LLM parameter that is captured during training? How do you model the information change when conditioned on different user queries?
- When adopting your framework during inference, how long does it take to execute your framework?

---

> ### Author Rebuttal · Authors · 2026-03-26
>
> Thank you for the thoughtful review and for recognizing the clarity of the formalization, the structure of the paper, and the importance of the problem. We appreciate the concerns regarding experimental scope, baselines, and interpretation, and would like to clarify the intended contribution and scope of the submission.
>
> First, our primary contribution is the Atomic Information Flow (AIF) framework itself: a global provenance view over an orchestration graph that models query, tool outputs, intermediate generations, and final response through semantic atoms. The compression section is intended as an initial downstream application, not the sole or final use case. The central claim of the paper is therefore not limited to document compression, but rather that AIF provides a principled way to reason about fine-grained attribution across a tool-augmented generation pipeline.
>
> Second, we agree that heterogeneous tools are an important direction. In this submission, however, we intentionally study an idealized retrieval-style setting so that the core attribution problem can be validated cleanly against document-grounded QA benchmarks. Our goal in this first paper is to test whether the atomic attribution machinery works in a controlled setting where support can be meaningfully evaluated, before extending to richer heterogeneous tools and more complex orchestration graphs. The current experiments should therefore be viewed as a controlled validation of the attribution framework, rather than a claim that the broader heterogeneous-tool setting is already fully addressed.
>
> Third, we agree that prior work on prompt and document compression is relevant and should have been positioned more explicitly as related downstream work. Our intent in the application section is not to claim that AIF supersedes the compression literature, but rather to show that AIF-derived signals can serve as useful supervision for context selection and compression. In that sense, Directed Information Compression is best understood as an initial demonstration that the flow signals produced by AIF are practically actionable.
>
> Fourth, on runtime and deployment, we intentionally focus this work on offline computation rather than online inference. The AIF pipeline is designed as an analysis and supervision framework that can be run offline to study provenance and generate training signals. The motivating compression use case follows this design: compute flow-derived supervision offline, then distill it into a lightweight policy model for efficient inference-time behavior.
>
> Fifth, regarding the concern that LLMs and network pipes are fundamentally different, our framework does not attempt to model all information stored in model parameters from pretraining. Instead, AIF models observable semantic information flow within a concrete orchestration instance. The unit of analysis is the semantic atom expressed in tool outputs, intermediate generations, and final responses. In other words, the framework is operational rather than mechanistic: it tracks what information is expressed and propagated in a particular run, not the entirety of latent knowledge inside model parameters.
>
> Finally, on the Table 2 results and the WikiMultihopQA drop, we agree this dataset highlights an important limitation of the current assignment procedure. In WikiMultihopQA, the method underperforms the baseline on both precision and recall, suggesting a broader attribution failure mode in noisy multi-hop settings rather than a simple precision-recall tradeoff. Our interpretation is that the current response-assignment prompt encourages over-inclusive grounding behavior by asking the model to identify all grounding facts that support each statement, including partial support and all facts used in an inference chain. In harder multi-hop examples, this can blur the boundary between necessary evidence and merely related evidence, leading both to noisy assignments and failure to isolate the minimal true support set.
>
> In summary, we agree that broader heterogeneous-tool evaluation, stronger comparison to compression baselines, and further efficiency improvements are important next steps. However, we believe the current submission already makes a meaningful contribution by introducing a precise framework for global provenance across an orchestration graph, validating that atomic attribution is competitive in controlled document-grounded settings, and showing that AIF-derived signals can support a useful downstream compression policy.

---

> > ### Author Rebuttal · Reviewer_aXiE · 2026-04-02
> >
> > I agree that the primary contribution is the Atomic Information Flow framework. However, since this is not a purely theoretical paper, solid and thorough experimental validation is essential for completeness.
> >
> > **W1.** The authors claim they intentionally studied an idealized retrieval-style setting, yet they use the word "tool" broadly throughout the paper. This creates a clear mismatch between the scope of the experiments and the broader AIF framework.  What exactly are all the tools discussed in the retrieval-style setting in the paper? What are the applicable tools in the AIF framework?
> >
> > **W2.** The author claimed the main contribution is in the retrieval-style setting, yet the experiments do not compare within it. How does your framework compare with existing retrieval-style literature? Without such comparisons, there is insufficient evidence to demonstrate the effectiveness of the framework within its own stated scope.
> >
> > **W3.** Prior work has shown that additional context can sometimes hurt model performance, as seen in in-context learning and context saturation [1][2]. Can the authors explain how AIF accounts for or addresses these scenarios, particularly given that context compression is a central application?
> >
> > [1] Peng, Hao, et al. "When does in-context learning fall short and why? a study on specification-heavy tasks." arXiv preprint arXiv:2311.08993 (2023).
> >
> > [2] Vladika, Juraj, and Florian Matthes. "On the influence of context size and model choice in retrieval-augmented generation systems." Findings of the Association for Computational Linguistics: NAACL 2025. 2025.
> >
> > **W4.** Distillation into a lightweight policy model is not discussed in the original submission. The effectiveness of this distillation process remains unclear. How effective is this distillation? How do you model the information loss during this distillation process?
> >
> > **Q1.** I accept this response. This should be incorporated into the paper.

---

### Official Review · Reviewer_T83e · 2026-03-12

**Soundness:** 2
**Presentation:** 3
**Significance:** 2
**Originality:** 2
**Overall Recommendation:** 3
**Confidence:** 4

**Summary:**

The paper proposes an "Atomic Information Flow" (AIF) model based on network flow theory to trace fine-grained attribution from tool outputs to final responses in RAG systems, and utilizes these attribution signals to fine-tune a Gemma3-4B model for document-level context compression.

**Compliance With Llm Reviewing Policy:**

Affirmed.

**Final Justification:**

Since I do not get any response until now, I decide to keep my score unchanged.

**Key Questions For Authors:**

1. Since the core methodology relies heavily on fine-grained "atomic" decomposition , why does the final compression strategy execute a highly coarse, document-level ablation? Doesn't this imply that the expensive atomic decomposition and matching processes are merely being used to generate pseudo-labels to train a standard document classifier?
2. Given that mature summary modules and prompt compression techniques already exist that achieve higher compression ratios with minimal information loss, how do the authors justify the superiority of this simple document-masking strategy? Why are there no comparisons with state-of-the-art summarization or prompt compression baselines in Table 3?
3. How would the AIF model's "min-cut" prediction accuracy hold up if the static dataset documents were replaced with outputs from actual dense retrievers (e.g., BGE or Contriever) that contain realistic noise and hard negatives?
4. Will the authors replace the rasterized plots (Figure 1 and Figure 5) with high-resolution, scalable vector graphics in a potential camera-ready version? Furthermore, can the authors provide a detailed system architecture diagram that explicitly outlines the concrete end-to-end workflow to replace the currently ambiguous Figure 2?

**Limitations:**

yes

**Strengths And Weaknesses:**

# Strengths
1. Conceptual Innovation: Introducing classical network flow theory (specifically, the max-flow min-cut theorem) to RAG system attribution and interpretability provides an interesting conceptual perspective.
2. Granular Evaluation Metric: The proposed atomic-level factual decomposition offers a more fine-grained approach to tracing information provenance compared to traditional document-level evaluation baselines.

# Weaknesses
1. Limited Novelty and Misaligned "Compression": Despite the heavy emphasis on "atomic" decomposition in the methodology, the actual downstream application—Directed Information Compression—simply performs coarse-grained, document-level binary filtering. The policy physically ablates entire documents deemed irrelevant. This reduces the approach to a standard document selector, lacking significant novelty, particularly since prior works like the BGM framework (https://aclanthology.org/2024.acl-long.562.pdf) have already introduced mature summarization modules that achieve superior context compression.
2. Poor Presentation and Bad Writing: The quality of the figures and the overall presentation falls short of top-tier conference standards. The core workflow diagram (Figure 2) is overly abstract, merely showing bubbles and colored dots, which completely fails to illustrate the actual end-to-end data pipeline or the concrete workflow mechanics. Additionally, critical plots, such as the performance trade-offs (Figure 1) and AUROC curves (Figure 5), are provided as blurry raster images (e.g., PNG/JPG) rather than scalable vector graphics. This severely impacts readability and reflects a lack of academic rigor.
3. Flawed Experimental Setup: The paper claims to address tool orchestration in RAG systems but entirely bypasses the "retrieval" phase in its experiments. Instead of employing real dense retrievers (like Contriever, E5, or BGE), the authors directly use static context documents provided by standard QA datasets as their "tool outputs". This highly idealized setup ignores real-world retrieval noise (e.g., highly similar hard negatives), making the claims regarding context compression and hallucination mitigation unconvincing for practical engineering environments.

---

> ### Author Rebuttal · Authors · 2026-03-26
>
> We thank the reviewer for the thoughtful feedback and for recognizing the conceptual value of the Atomic Information Flow (AIF) framework, particularly the network-flow perspective and the fine-grained provenance formulation. We appreciate the concerns regarding the granularity of the downstream application, the experimental scope, and the clarity of presentation, and we would like to clarify the intended contribution of the paper.
>
> Our primary contribution is the AIF framework itself: a general formulation for tracing fine-grained information provenance in tool-augmented generation by modeling the flow of atomic units of information from tool outputs to final responses. The document-level compression experiment is intended as a first downstream validation of whether these attribution signals are useful, not as the full representational limit of the framework. We agree that document-level filtering is a coarse application relative to the underlying atomic formulation. This choice was intentional. Existing multi-hop QA benchmarks provide useful supervision in the form of supporting facts or evidence chains, but they do not directly supervise the broader atom-level provenance problem that AIF is designed to model. Our goal in this paper is therefore to validate that the core flow-assignment procedure is technically feasible and yields actionable supervision in a benchmark-consistent setting before moving to finer-grained downstream tasks such as span-level or atom-level prediction.
>
> For the same reason, we chose document-level exact-match outcomes as a clean proxy target for benchmarking whether AIF-derived attribution signals are useful across multiple benchmarks. In this sense, the atomic decomposition and matching are not introduced merely to train a standard document classifier. Rather, they are the core mechanism that generates structured supervision for a more general provenance problem, while document-level filtering serves as a controlled first application. We agree that richer downstream instantiations are important future work, but we view them as natural extensions of the framework rather than prerequisites for establishing its core validity.
>
> Regarding comparisons to summarization or prompt compression approaches such as BGM, we agree these are closely related and important baselines for future work. However, we respectfully note that they address a partially different objective. Summarization-based methods compress context by rewriting or abstracting it, whereas our focus here is on identifying which upstream information is actually necessary for preserving answer correctness under an explicit attribution model. The document-masking strategy is therefore not intended to claim superiority over all compression paradigms, but to provide a simple and controlled downstream setting in which the usefulness of AIF-derived signals can be directly evaluated. We agree that broader empirical comparisons would strengthen the compression application, and we view this as a valuable direction for future work.
>
> On the experimental setup, we intentionally scope the current evaluation to the generation component rather than the full retrieval pipeline. This was done to isolate attribution quality under relatively clean support conditions and to validate the core AIF assignment procedure before introducing additional confounds from realistic retrieval noise. We agree with the reviewer that real-world dense retrievers introduce hard negatives, conflicting evidence, and claim-resolution challenges that are not fully captured by static benchmark contexts. We see this as an important next step for extending the framework, rather than as a contradiction of the current results. The present paper is meant to establish feasibility in a controlled setting first, with the expectation that future work can extend AIF to non-idealized retrieval scenarios.
>
> Finally, we agree with the presentation comments. In a camera-ready version, we would replace the rasterized plots with high-resolution vector graphics and revise Figure 2 to provide a clearer end-to-end system diagram that explicitly illustrates the concrete workflow.
>
> We hope this clarifies the intended scope of the paper. The goal is to introduce AIF as a general framework for fine-grained provenance in tool-based generation and to demonstrate, through a deliberately simple downstream task, that its attribution signals are technically feasible and practically useful. We appreciate the reviewer’s comments and believe they point directly toward the most important directions for expanding the framework in future work.

---

### Official Review · Reviewer_Bg6p · 2026-03-12

**Soundness:** 2
**Presentation:** 3
**Significance:** 2
**Originality:** 2
**Overall Recommendation:** 3
**Confidence:** 2

**Summary:**

The paper proposes Atomic Information Flow (AIF), which models RAG pipelines as directed flow networks. The idea is to break tool outputs into "atoms" (minimal semantic units), inject relevance signals, and then trace which atoms end up in the final response. They define several flow heuristics for attribution and use a max-flow/min-cut formulation to do context compression. The compression results on HotPotQA are honestly pretty impressive — Gemma3-4B jumps from ~55% to ~83% with massive token savings.

**Compliance With Llm Reviewing Policy:**

Affirmed.

**Key Questions For Authors:**

1. For the compression results specifically, I kept wondering: how much of this is the flow-based signal vs. just having any decent relevance signal? The paper doesn't compare against BM25 selection or embedding-based retrieval, which would be cheap baselines. Even something as simple as training on oracle paragraph labels would help calibrate expectations. Without these, it's hard to tell if the max-flow formulation is doing real work or if you'd get similar gains from simpler approaches.

2. Have you tried running AIF on any system with genuinely different tool types (search + calculator, or retrieval + code execution)? Even a small-scale qualitative analysis would strengthen the multi-tool claims significantly.

**Limitations:**

See weakness above.

**Strengths And Weaknesses:**

Strengths:

- The compression experiment (Table 3) is the strongest part of the paper. The accuracy-efficiency tradeoff shown in Figure 1 makes a convincing case that there's practical value here, especially if you care about inference cost.

- The network flow framing is a neat way to think about attribution. It's not entirely new (there's been work on information flow in attention mechanisms, e.g., Abnar & Zuidema 2020), but applying it at the semantic atom level is a reasonable contribution.

Weakness:

- My main concern is the gap between the paper's claims and the experimental setup. The title and intro talk about "tool attributions in RAG systems" and "complex multi-agent architectures," but every experiment just takes QA dataset passages and relabels them as "tool calls." That's... not really multi-tool RAG. Real tool-augmented systems involve heterogeneous outputs (API JSON, database rows, code execution results), sequential dependencies, etc. I don't doubt the framework could extend there, but the current experiments don't show it, and I think the paper would be stronger if the authors were more upfront about this scope.

- The evaluation setup worries me a bit. GPT4.1 handles atom matching in Algorithm 3 and also serves as the LLM judge (Prompt F), while GPT5-Nano does the decomposition. These are related model families, and if they share biases around, say, lexical overlap or paraphrase detection, the metrics could be artificially high. The 50-instance human eval on just HotPotQA isn't really enough to rule this out — I'd want to see at least a couple hundred instances across multiple datasets, ideally with inter-annotator agreement reported.

---

> ### Author Rebuttal · Authors · 2026-03-25
>
> Thank you for the thoughtful review and for recognizing the strength of the compression results and the practical value of the reported accuracy-efficiency tradeoff. We appreciate the concerns around scope, evaluation setup, and baseline calibration, and we would like to clarify several factual points that directly affect this assessment.
>
> First, regarding baseline calibration for compression: the submission already includes a cheap lexical relevance baseline in Figure 1 (“Standard BM25 Frontier”), which shows the accuracy-efficiency tradeoff achieved by BM25-based selection across token budgets. The AIF-trained Gemma3-4B compressor improves substantially beyond this frontier and approaches the performance of Qwen3-4B at the same model scale. We agree this comparison should have been called out more explicitly in the main text rather than relying primarily on the figure. However, the intended comparison against a simple relevance-based selector is already present in the submission, and the observed gains are not solely relative to the uncompressed baseline.
>
> Second, on evaluation bias: we agree that using GPT-5-Nano and GPT-4.1 introduces a potential concern because they are both OpenAI models and may share tendencies in paraphrase or lexical-overlap judgments. This is a fair limitation, and we will state it more explicitly. At the same time, the paper contains two distinct evaluations that should be separated more clearly. In Table 2, attribution quality is measured using precision/recall/F1 against ground-truth retrieval document titles from HotPotQA. In Table 3, compression is evaluated through downstream answer correctness, with GPT-4.1 used only as the final judge. The 50-instance human evaluation is intended as a sanity check on attribution quality, not as the sole basis for validation, and we will revise the paper to make that role clearer.
>
> Third, on scope: we agree that the current experiments are conducted in a controlled retrieval-style setting rather than on a fully heterogeneous tool ecosystem such as search+calculator or retrieval+code execution. Our intention in this submission is to introduce and validate the core AIF mechanism—atom-level attribution over tool-to-response edges—in a setting where attribution quality and compression utility can be measured cleanly. We agree that the current empirical evidence does not yet justify broad claims about all complex multi-tool or multi-agent architectures, and we will revise the title/introduction framing to make this boundary explicit. We view broader validation on heterogeneous tool outputs as an important next step rather than a claim established by the current experiments.
>
> More broadly, the goal of this paper is not to claim that the present decomposition/assignment pipeline is the definitive implementation of AIF, but to introduce the framework and validate it with a first concrete instantiation. That initial instantiation is likely not optimal along several axes, including decomposition model choice, attribution model choice, and atom granularity. We will make this clearer in revision. Our claim is therefore not that every implementation detail is final, but that even an initial implementation already yields strong enough attribution and compression results to demonstrate that the AIF perspective is a promising and practically useful direction for further work.
>
> We appreciate the reviewer’s suggestions and will revise the paper to: (1) foreground the BM25 comparison in the main text, (2) clarify the distinct roles of the attribution and compression evaluations, and (3) narrow the framing so that the empirical scope matches the claims more precisely.

---

### Official Review · Reviewer_71B5 · 2026-03-13

**Soundness:** 3
**Presentation:** 3
**Significance:** 3
**Originality:** 3
**Overall Recommendation:** 3
**Confidence:** 3

**Summary:**

This paper proposes Atomic Information Flow (AIF), a graph/network-flow view of tool-based RAG systems. In this view, each tool/LLM call is a node and "atoms" (small, self-contained units of information) flow from a query super-source to a response super-sink. AIF makes provenance work by (i) breaking down tool outputs into atomic facts, (ii) linking response atoms back to tool atoms to create fine-grained tool attributions, and (iii) combining these into flow-derived metrics (like groundedness or tool consumption) for debugging and analysis.

The authors present Directed Information Compression via min-cut as an application. They utilize AIF-derived "minimum cut" style signals to train a lightweight Gemma3-4B policy model that forecasts which tool outputs can be eliminated while maintaining answer accuracy. They report significant improvements over the base 4B compressor on multi-hop QA benchmarks, with HotpotQA accuracy increasing from 54.7% to 82.71% while achieving comparable token reduction.

**Compliance With Llm Reviewing Policy:**

Affirmed.

**Final Justification:**

I did my best to quickly highlight the concerns that remained post-rebuttal. Since the discussion period is ending now and follow-up comments were not addressed, I am leaning towards maintaining my initial score.

**Key Questions For Authors:**

What is the extra value of the "network flow / min-cut"? Can you show an ablation where supervision from simpler baselines (like retrieval scores, citation overlap, or ALCE-style signals) matches or doesn't match AIF-derived labels for training a compressor?

**Limitations:**

yes

**Strengths And Weaknesses:**

Strengths:

1. Clear problem motivation and framing (significance). Tool-based RAG systems are becoming increasingly complex, and determining which tool outputs affect the final answer is important for debugging, safety, and efficiency. The paper delineates this gap explicitly and establishes AIF as a universal mechanism for tool attribution within these pipelines.

2. The idea of modeling provenance as a flow network over atomic semantic units is a good mix of (a) factual decomposition and (b) attribution/grounding. It is also based on max-flow/min-cut ideas. The components look like earlier "fact decomposition + matching" methods, but the network-flow lens and min-cut application form a new, coherent conception.

3. AIF-derived labels seem to allow a small compressor to approximate the performance of significantly larger counterparts. Table 3 shows that the Gemma3-4B compressor consistently achieves large gains across multiple datasets while reducing the number of tokens by a large amount.

4. The paper reports a human evaluation sanity check for the decomposition and attribution stages on HotpotQA with high agreement rates.

Weaknesses:

1. The comparison of ALCE and AIF tool-attribution correctness shows improvements on some datasets/segments but also regressions (for example, on Wiki Multihop QA, "TRUE" precision/F1 seems worse for AIF than the baseline in Table 2). This necessitates a more detailed examination of the circumstances under which AIF is beneficial versus detrimental, and whether the matching and labeling processes engender systemic bias.

2. I would like to see more controlled ablations around atom granularity choices, matching strategy, sensitivity to relevance thresholds, and dependence on the answer generator.

3. Experiments and post-training focus mostly on tool-to-response ("Generation") edges, and the "Query-to-Tool / Retrieval" part of the graph is left for later work. This is a good first step, but it only partially proves the framework's claim of "end-to-end RAG orchestration".

---

> ### Author Rebuttal · Authors · 2026-03-26
>
> We thank the reviewer for the thoughtful assessment and for recognizing the paper’s motivation, originality, and the strength of the compression results.
>
> On the WikiMultihopQA gap in Table 2, we agree this dataset highlights an important limitation of the current assignment procedure. In WikiMultihopQA, the method underperforms the baseline on both precision and recall, suggesting a broader attribution failure mode in noisy multi-hop settings rather than a simple precision-recall tradeoff. Our interpretation is that the current response-assignment prompt encourages over-inclusive grounding behavior by asking the model to identify all grounding facts that support each statement, including partial support and all facts used in an inference chain. In harder multi-hop examples, this can blur the boundary between necessary evidence and merely related evidence, leading both to noisy assignments and failure to isolate the minimal true support set. We also do want to note that we present the side by side comparison with the ALCE benchmark primarily as a sanity check of the attribution model and that we can achieve comparable performance, while expanding the broader heuristic set we can support. We agree that improving the underlying attribution mechanism is an essential next step to expand upon this work.
>
> We also agree that the current experiments focus on Tool-to-Response (“Generation”) edges rather than fully validating Query-to-Tool (“Retrieval”) edges. This was a deliberate scoping choice. Our goal in this submission was to first validate that atom-level attribution with language models is feasible and useful on the generation side, where response grounding and hallucination analysis are immediate practical problems. We see retrieval-edge modeling as a natural extension of the same framework rather than a separate claim.
>
> Regarding the extra value of the network-flow / min-cut lens: our main claim is not only that AIF produces attribution labels, but that it turns local atomic attributions into structured flow quantities over the orchestration graph. This graph-level view enables downstream optimization objectives that simpler attribution signals do not directly provide. In particular, the directed information compression result relies on supervision derived from the flow/min-cut perspective, and this signal materially improves the Gemma3-4B compressor across datasets. We therefore see the main evidence for the value of the flow lens not only in attribution metrics, but in its ability to generate useful training signals for compression.
>
> More broadly, we agree that controlled ablations over atom granularity, matching choices, thresholding, and generator dependence would strengthen the paper. We view these as important next steps in optimizing the proposed framework. The central contribution of this work is to introduce Atomic Information Flow as a unified modeling lens for provenance in tool-augmented RAG systems, and to show that even this first implementation already supports a strong downstream application in context compression.

---

> > ### Author Rebuttal · Reviewer_71B5 · 2026-04-01
> >
> > I thank the authors for their responses.
> >
> > **W1.** If the cause is the prompt design, a straightforward ablation should recover precision on WikiMultihopQA. Was this attempted? Without it, the explanation remains a hypothesis.
> >
> > **W2.** I understand the framing. However, without ablations at the atom level, matching strategy, or relevance thresholds, it is not possible to assess sensitivity to these design choices. The work would gain from deeper experimental grounding.
> >
> > **W3.** I accept this response.

---

### Decision · Program_Chairs · 2026-04-30

**Decision:**

Reject

**Comment:**

The submission introduces an interesting conceptual framework by applying network flow theory to RAG systems. Reviewers unanimously noted that despite the fine-grained atomic theoretical framing, the actual downstream application reduces to a standard, coarse document selector that fails to differentiate itself conceptually or experimentally from existing prompt compression baselines. Furthermore, the heavily empirical evaluation relies on an idealized setup that bypasses real-world retrieval noise and tool environments. The work requires deeper conceptual development and more rigorous benchmarking against state-of-the-art baselines.